Does afforestation deteriorate haze pollution in Beijing–Tianjin–Hebei (BTH), China?
Xin Long[1, 3], Naifang Bei[2], Jiarui Wu[1], Xia Li[1], Tian Feng[1], Li Xing[1], Shuyu Zhao[1], Junji Cao[1], Xuexi Tie[1, 4],
Zhisheng An[1], and Guohui Li[1*]
[1]Key Lab of Aerosol Chemistry & Physics, SKLLQG, Institute of Earth Environment, Chinese Academy of
Sciences, Xi'an, 710061, China
[2]Institute of Global Environmental Change, Xi'an Jiaotong University, Xi'an, 710049, China
[3]Joint Center for Global Change Studies (JCGCS), Beijing 100875, China
[4]National Center for Atmospheric Research, Boulder, CO, 80303, USA
[*]Correspondence to: Guohui Li (ligh@ieecas.cn)
**Abstract:** Although aggressive emission control strategies have been implemented recently
in the Beijing-Tianjin-Hebei area (BTH), China, pervasive and persistent haze still frequently
engulfs the region during wintertime. Afforestation in BTH, primarily concentrated in the
Taihang and Yanshan Mountains, has constituted one of the controversial factors exacerbating
the haze pollution due to its slowdown of the surface wind speed. We report here an
increasing trend of forest cover in BTH during 2001-2013 based on long-term satellite
measurements and the impact of the afforestation on the fine particles ($PM_{2.5}$) level.
Simulations using the Weather Research and Forecast model with chemistry reveal that the
afforestation in BTH since 2001 generally deteriorates the haze pollution in BTH to some
degree, enhancing $PM_{2.5}$ concentrations by up to 6% on average. Complete afforestation or
deforestation in the Taihang and Yanshan Mountains would increase or decrease the $PM_{2.5}$
level within 15% in BTH. Our model results also suggest that implementing a large
ventilation corridor system would not be effective or beneficial to mitigate the haze pollution
in Beijing.

## 1  Introduction

Heavy haze with extremely high levels of fine particles ($PM_{2.5}$), caused by rapid growth of industrialization, urbanization, and transportation, frequently covers Northern China during wintertime, particularly in the Beijing–Tianjin–Hebei area (BTH). The haze pollution in BTH remarkably impairs visibility and potentially causes severe health defects (Lim et al., 2013; Wang and Hao, 2012). The Chinese State Council has issued the 'Air Pollution Prevention and Control Action Plan' (APPCAP) in September 2013 with the aim of improving China's air quality within five years and reducing $PM_{2.5}$ by up to 25% by 2017 relative to 2012 levels. Although aggressive emission control strategies have been undertaken since the initiation and implementation of the Action Plan, widespread and persistent haze still often engulfs BTH.

Aside from emissions, meteorological conditions play a key role in the haze pollution, affecting the formation, transformation, diffusion, transport, and removal of $PM_{2.5}$ in the atmosphere (Bei et al., 2012; 2017). Multifarious measurements have provided cumulative evidence that the widespread slowdown of surface wind speeds has occurred globally and in China since 1980s (Chen et al., 2013; McVicar et al., 2012), which facilitates the pollutant accumulation to deteriorate air quality (Zhao et al., 2013; Sun et al., 2016). An increase in the surface roughness induced by increased vegetative biomass has been proposed to be responsible for the surface-level stilling to some degree (Wu et al., 2016b; Vautard et al., 2010). Consequently, a debate has been circulated in China on whether the afforestation program contributes importantly to the haze formation in BTH (China foresty network, 2016a, 2017).

Deforestation and its potentials to severe droughts and massive floods raised serious concerns in China since 1970s, fostering the largest afforestation project in the world (Liu et al., 2008). Six key afforestation programs have been implemented since 2001 and "the Green

Great Wall" of China has been established in Northern China (Duan et al., 2011). A
remarkable forest growth has been reported in the northwest of BTH from 2000 to 2010 (Li
et al., 2016), which has the potential to increase the surface roughness and decrease the
surface wind speed (Wu et al., 2016a; Bichet et al., 2012), and could potentially aggravate the
haze pollution. In addition, previous studies have shown that the afforestation is beneficial for
the atmosphere to remove $O_3$, $NO_x$, $SO_2$, and $PM_{2.5}$ through the dry deposition process
(Zhang et al., 2015; 2017; Huang et al., 2016). Hence, a large artificial ventilation corridor
system has been proposed, highly anticipated to ventilate Beijing (China forestry network,
2014, 2016b, c).
In the present study, we report an analysis of long-term satellite measurements of the
land cover change in BTH and quantitatively evaluate the impacts of the afforestation on the
haze pollution in BTH using the WRF-CHEM model. We have further evaluated the effect of
the proposed large artificial ventilation corridor system on the haze mitigation in Beijing. The
model configuration and methodology are provided in Section 2. Data analysis and model
results are presented in Section 3, and conclusions are given in Section 4.

**2    Data, model and methodology**
**2.1  MODIS data**
The data utilized in the study are the annual land cover product, MCD12Q1, derived
from the Terra- and Aqua- Moderate Resolution Imaging Spectroradiometer (MODIS)
observations since 2001 (Friedl et al., 2002). The product has been widely used in studies of
atmospheric science, hydrology, ecology, and land change science (Gerten et al., 2004;
Guenther, 2006; Reichstein et al., 2007; Turner et al., 2007). Wu et al., (2008) have compared
four global land cover datasets across China, concluding that the MODIS land cover product
is the most representative over China with the minimal bias from the China's National Land

Cover Dataset. The MCD12Q1 (Version 5.1) IGBP (International Geosphere Biosphere Programme) scheme with a spatial resolution of 500 m is utilized to explore the variability of the land cover fraction (LCF) from 2001 to 2013 in BTH and assimilated into the WRF-CHEM model. The high-resolution land cover product is generated using a supervised classification algorithm in conjunction with a revised database of high quality land cover training sites (Friedl et al., 2002). The accuracy of the IGBP layer of MCD12Q1 is estimated to be 72.3-77.4% globally, with a 95% confidence interval on the estimate of 72.3-77.4% (Friedl et al., 2002; Friedl et al., 2010). Great efforts have been made to evaluate the accuracies of the global land cover datasets over China. The overall accuracy of MCD12Q1 in China is estimated to be 55.9-68.9% (Bai et al., 2015; Yang et al., 2017), which could be increased to about 70% when ignoring the differences of five forests.

## 2.2 WRF-CHEM model and configurations

We use a specific version of the WRF-CHEM model (Grell et al., 2005) to investigate the impacts of the afforestation on the haze pollution in BTH. The model includes a new flexible gas phase chemical module and the CMAQ/Models3 aerosol module developed by US EPA (Binkowski and Roselle, 2003). The wet deposition of chemical species follows the CMAQ method. The dry deposition parameterization follows Wesely (Wesely, 1989), and the dry deposition velocitiy of aerosols and trace gases is calculated as a function of the local meteorology and land use. The photolysis rates are calculated by FTUV (Fast Radiation Transfer Model) (Li et al., 2005). The inorganic aerosols are predicted using ISORROPIA Version 1.7 (http://nenes.eas.gatech.edu/ISORROPIA/) (Nenes et al., 1998). The secondary organic aerosol (SOA) is predicted using a non-traditional SOA module, including the VBS (volatility basis-set) modeling approach and SOA contributions from glyoxal and methylglyoxal. Detailed information about the WRF-CHEM model can be found in previous studies (Li et al., 2010; Li et al., 2011a, b; Li et al., 2012).

High PM$_{2.5}$ pollution episodes from 1 December 2013 to 31 January 2014 in the North
China Plain (NCP) have been simulated using the WRF-CHEM model. The model simulation
domain is shown in Figure 1, and detailed model configurations can be found in Table 1. The
chemical initial and boundary conditions are interpolated from the 6h output of MOZART-4
(Emmons et al., 2010; Horowitz et al., 2013). MOZART-4 is driven by meteorology fields
from the NASA GMAO GEOS-5 model, using anthropogenic emissions based on David
Streets' inventory (Streets et al., 2006) and fire emissions from FINN-v1 (Widinmyer et al.,
2011). The model has been evaluated comprehensively with several sets of observations,
reproducing well tropospheric chemical composition (Emmons et al., 2010). The model
results have been successfully and widely used as the initial and lateral boundary conditions
for chemical transport models. The anthropogenic emission inventory used in the present
study is developed by Zhang et al. (2009), with the base year of 2013, including contributions
from agriculture, industry, power, residential and transportation sources. Figure S1 shows the
emission distribution of OC, VOCs, NO$_x$, and SO$_2$ in the simulation domain. The high
emissions of OC, VOCs, NOx, and SO$_2$ are generally concentrated in the plain region of
BTH and Shandong province, the downwind area of afforestation.
The hourly near-surface CO, SO$_2$, NO$_2$, O$_3$, and PM$_{2.5}$ mass concentrations released by
the China's Ministry of Environmental Protection are used to validate the model simulations
and accessible from the website http://www.aqistudy.cn/.
We use the normalized mean bias (*NMB*), the index of agreement (*IOA*), and the
correlation coefficient (*R*) to assess the WRF-CHEM model performance in simulating air
pollutants against measurements.
$$NMB = \frac{\sum_{i=1}^{N}(P_i - O_i)}{\sum_{i=1}^{N} O_i} \tag{1}$$
$$IOA = 1 - \frac{\sum_{i=1}^{N}(P_i - O_i)^2}{\sum_{i=1}^{N}(|P_i - \bar{O}| + |O_i - \bar{O}|)^2} \tag{2}$$
$$R = \frac{\sum_{i=1}^{N}(P_i - \bar{P})(O_i - \bar{O})}{[\sum_{i=1}^{N}(P_i - \bar{P})^2 \sum_{i=1}^{N}(O_i - \bar{O})^2]^{\frac{1}{2}}}$$ (3)
Where $P_i$ and $O_i$ are the calculated and observed air pollutant concentrations, respectively.
$N$ is the total number of the predictions used for comparisons, and $\bar{P}$ and $\bar{O}$ represent the
average of predictions and observations, respectively. The *IOA* ranges from 0 to 1, with 1
showing perfect agreement of the prediction with the observation. The *R* ranges from -1 to 1,
with 1 implicating perfect spatial consistency of observations and predictions.
**2.3 MCD12Q1 data assimilation to the WRF-CHEM model**
The IGBP layer in MCD12Q1 is suitable for the WRF-CHEM IGBP land cover scheme,
which consist of 11 natural vegetation classes, 3 developed and mosaicked land classes, and
three non-vegetated land classes. Table S1 displays the comparison of land cover
classification between the WRF-CHEM model and MCD12Q1. We use the gridded LCF of
each category to assimilate the MCD12Q1 satellite data to the WRF-CHEM model.
$$LCF_{i,j,k} = \frac{Area_{i,j,k}}{Area_{i,j}}$$ (4)
Where i and j are grid cell indices of the WRF-CHEM model domain, $Area_{i,j,k}$ stands for the
total area of each land cover category k within grid cell (i, j), and $Area_{i,j}$ is the area of grid
cell (i, j). The $LCF_{i,j,k}$ ranges from 0 to 1.
To evaluate the afforestation impacts on the haze pollution in BTH, we have used and
modified the coupled unified Noah land-surface model (LSM), which was developed based
on Oregon State University LSM (Chen and Dudhia, 2001). The Noah is able to reasonably
reproduce the observed diurnal variation of sensible heat fluxes and surface skin temperature.
Also, it is capable of capturing the diurnal and seasonal evolution in evaporation and soil
moisture (Chen et al., 1996). Despite some remaining issues, the Noah has been chosen for
further refinement and implementation in NCEP regional and global coupled weather and
climate models because of its relative simplicity and adequate performance (Mitchell, 2005).
The surface roughness length (SFz0) in Noah is calculated based on the dominant land cover
category (https://ral.ucar.edu/solutions/products/unified-noah-lsm).
$$SFz0 = \begin{cases} SFz0_{min}, & G_T \leq G_{min} \\ (1 - G_f) * SFz0_{min} + G_f * SFz0_{max}, & G_{min} \leq G_T \leq G_{max} \\ SFz0_{max}, & G_T \gg G_{max} \end{cases} \quad (5)$$

Where $SFz0_{min}$ and $SFz0_{max}$ are the minimal and maximum SFz0 for each category. $G_f$ is
the area fractional coverage of green vegetation, and $G_T$, $G_{min}$ and $G_{max}$ are the threshold,
minimal, and maximal value of $G_f$, respectively. $G_T$, $SFz0_{min}$ and $SFz0_{max}$ are listed in
Table S2.
In order to more precisely simulate surface stress within the sub-grid scale in
heterogeneous terrain, the effective roughness length has been extensively studied, especially
in the 1990s. Claussen (1990) has defined the effective roughness length as a value of the
area average of the roughness length in heterogeneous terrain. The effective roughness length
relies upon the blending height (Wieringa, 1986; Mason, 1988; Wood and Mason, 1991;
Philip, 1996; Mahrt, 1996), at which the flow is approximately in equilibrium with
underlying surface conditions and independent of horizontal position (Ma and Daggupary,
1998). We have modified the Noah SFz0 calculation using the spatial average of the
vegetation roughness length.
$$SFz0 = \sum_k LCF_k * SFz0_k \quad (6)$$

*SFz0$_k$* denotes the gridded area fraction of land cover category *k*, and calculated by Eq. (5).

**3    Results and Discussions**
**3.1  Land cover change in BTH**
The land cover in BTH and Beijing exhibits appreciable variation from 2001 to 2013
(Figure 2 and Table 2). In BTH, forests and croplands have increased by 7.2% and 1.9%, while
shrublands and grasslands/savannas have decreased by 3.9% and 5.1%, respectively. In Beijing,
forests have increased by 14.9%, while shrublands have decreased by 12.6%. Apparently, the
forest LCF has increased substantially in western and northern BTH, concentrated in the
Taihang and Yanshan Mountains, with an increase up to 50%. This result is consistent with
the previous study of Li et al. (2016), which has reported a remarkable forest growth in the
northwest of NCP from 2000 to 2010. As such, a "Green Great Wall" has been established
(Figure 2a), which has reportedly protected the southeastern BTH from the dust pollution
(Liu et al., 2008; Duan et al., 2011; Parungo et al., 2013). The land cover change, particularly
the evident forest growth, is primarily attributed to the China's national afforestation programs
aiming to increase the forest coverage and to conserve soil and water, including the Grain for
Green Project, the Three Norths Shelter Forests System Project (Phase IV), and the Natural
Forest Conservation Program (Yin et al., 2010; Cao et al., 2011).
**3.2 Model performance**
We have first assimilated into the WRF-CHEM model the MCD12Q1 product of 2013
and performed the numerical simulation of haze pollution episodes from 1 December 2013 to
31 January 2014. For the discussion convenience, we have defined the simulation with the
2013 land cover as the reference case (hereafter referred to as REF case), and results from the
reference simulation are compared to observations in BTH.
Considering the key role of meteorological fields in determining the formation,
transformation, diffusion, transport, and removal of the air pollutants (Bei et al., 2017),
Figure S2 presents the comparison of the simulated wind speed and direction, and planetary
boundary layer height with the reanalysis data from ECMWF (European Centre for
Medium-range Weather Forecasts) at monitoring sites. The predicted temporal variations of
the three meteorological parameters are generally in agreement with the reanalysis data, with
the *IOAs* exceeding 0.80, and the absolute *NMB* less than 25%.
Figure 3 presents the calculated and observed temporal profiles of near-surface air
pollutants concentrations averaged at monitoring sites in BTH during the simulation period,
including $PM_{2.5}$, $O_3$, $NO_2$, $SO_2$, and CO. The WRF-CHEM model generally reproduces the
haze pollution episodes well, e.g. all the haze events during the period are captured
successfully (Figure 3a), with an *IOA* of 0.90 and a *NMB* of 2.1% for $PM_{2.5}$ mass
concentrations. The model reasonably yields $O_3$ variations compared to observations, with an
*IOA* of 0.80, but underestimates $O_3$ concentrations, with a *NMB* of -15.9% (Figure 3b). In
winter, the insolation is weak in the north of China, unfavorable for the $O_3$ photochemical
production, so the $O_3$ level is substantially influenced by the boundary conditions. Hence, one
of possible reasons for the $O_3$ underestimation might be from the uncertainty in the $O_3$
boundary conditions. The simulated temporal variations of $NO_2$ mass concentrations are well
consistent with the observation, and the *IOA* and *NMB* are 0.91 and 0.6%, respectively. The
$SO_2$ and CO temporal variations are also reasonably replicated against observations, with
*IOAs* of 0.82 and 0.84, respectively.

Figure 4 shows the spatial comparison of calculated and observed $PM_{2.5}$ concentrations.

Generally, the average predicted $PM_{2.5}$ spatial patterns agree well with the observations at the
monitoring sites in BTH during the whole period (Figure 4b) and each month (Figures 4c and
4d), with *R*s exceeding 0.85, indicating good agreement of simulations with observations.
The observed $PM_{2.5}$ concentrations frequently exceed 150 μg m$^{-3}$ in BTH, showing the
frequent occurrence of heavy haze pollution events. The model generally yields the observed
high $PM_{2.5}$ concentrations in BTH and their surrounding areas, although the model
underestimation or overestimation still exists. Additionally, compared to observations, the
model performs also well in simulating the spatial pattern of haze episodes with various
time-scales ranging from 8 to 16 days (Figure S3).

The good agreements of the simulated mass concentrations of air pollutants with

observations at monitoring sites in BTH show that the emission inventory used in present
study and simulated wind fields are generally reasonable, providing a reliable base for the
further assessment. It is worth noting that, although the predicted meteorological parameters
are generally consistent with the reanalysis data from ECMWF at monitoring sites, other
factors still affect the meteorological field simulations and cause biases to compensate some
of the deficiencies of the WRF-CHEM model, such as overestimation of surface wind speeds.
**3.3   Effect of afforestation on haze pollution in BTH**
Change in the land cover alters the surface roughness height (SFz0) that plays an
important role in determining the surface level wind speed and energy exchange between the
atmosphere and the land surface. Numerous studies have demonstrated that increasing SFz0
tends to decelerate the surface wind (Wu et al., 2016a, b), obstructing the dispersion of air
pollutants (Sun et al., 2016; Zhao et al., 2013; Tie et al., 2015). In order to evaluate the
impact of the afforestation induced SFz0 change and resultant dynamical change (e.g., wind
field) on the haze formation, a sensitivity experiment is designed, in which the MCD12Q1
product of 2001 is assimilated into the WRF-CHEM model to represent the land cover
situations before the afforestation (hereafter referred to as SEN-AFF case).
Figures 5a and 5b display the SFz0 change and its correlation with forest LCF change
from 2001 to 2013, respectively. The land cover change considerably alters the SFz0,
particularly in the afforestation area, with a SFz0 increase ranging from 0.1 to 0.3 m.
Apparently, the SFz0 exhibits a distinct increasing trend in western and northern BTH,
concentrated in the Taihang and Yanshan Mountains, which is well consistent with the
increase of the forest LCF. The SFz0 change is highly correlated with the forest LCF change,
with a correlation coefficient of 0.91. Generally, the SFz0 is mainly dependent upon the LCF
(Equation 6), and sensitive to the forest change (Table S2). Therefore, afforestation
constitutes the most important factor for the increase in the SFz0 in BTH.
It is worth noting that Jiménez and Dudhia (2012) have point out that there still exist
large uncertainties in parameterizing the air land interaction over complex terrain. Besides the
vegetation effect on the roughness length, drag of subgrid features of topography need to be
considered. The parameterization of orographic flow over complex terrain is a challengeable
problem at the mesoscale numerical simulation. In early versions of the WRF model, a large
bias in predicting surface winds over complex terrain has occurred due to the drag exerted by
unresolved topography (Cheng and Steenburgh, 2005). Great efforts have been made to
improve the simulation of orographic flow over complex terrain. The new parameterization
scheme introduced in the WRF model since version 3.4.1 has corrected this high wind speed
bias over plains and valleys (Mughal et al., 2017), and also corrected the low wind speed bias
found over the mountains and hills (Jiménez and Dudhia, 2012).
Figures 5c and 5d illustrate the influence of the afforestation on the surface $PM_{2.5}$ and
wind field averaged during the simulation period (defined as (REF − SEN-AFF)). The
prevailing westerly or northerly wind is decelerated in the western and northwestern BTH
due to the increased SFz0 caused by the afforestation, with the wind speed decrease ranging
from 0.3~1.5 m s$^{-1}$. The afforestation tends to deteriorate the haze pollution in BTH,
particularly in the downwind area of the afforestation, with the period average $PM_{2.5}$
enhancement reaching about 6~15 μg m$^{-3}$, or 3~6%. The $PM_{2.5}$ enhancement in Beijing is the
most evident, corresponding to the rapid growth of forests in the west and in/on the north of
Beijing. Furthermore, during each episode, the afforestation generally tends to deteriorate the
haze pollution in BTH, enhancing the $PM_{2.5}$ concentration by about up to 3~6%, particularly
in the downwind area of the afforestation (Figure S4). On average, the difference of the
simulated air pollutants and meteorological parameters between the REF and SEN-AFF case
is not significant (Figures 5c and 5d).
The occurrence of heavy haze pollution in BTH is generally associated with the
weakening of northerly or northwesterly winds, which facilitates the accumulation of air
pollutants in BTH. The afforestation in the western and northwestern BTH increases SFz0,
further decelerating northerly or northwesterly winds and deteriorating the haze pollution.
However, the afforestation only plays a marginal role in worsening the haze pollution, and
does not constitute the main cause for the heavy haze formation.

Apparently, during the haze development, when the northerly or northwesterly wind is

weak or becomes calm, the SFz0 increase due to the afforestation contributes negligibly to
the haze deterioration in BTH. However, once the northerly or northwesterly wind
commences to strengthen but is not strong enough to evacuate the air pollutants in BTH, the
SFz0 increase would play an appreciable role in sustaining high $PM_{2.5}$ levels in the
downwind area of the afforestation. Figure 6 presents the $PM_{2.5}$ contribution of the
afforestation during the occurrence of a northerly gust on January 18, 2014. The intensified
northerly wind cleanses the northern BTH, but the haze pollution is still very severe in the
southern BTH. The afforestation considerably elevates the $PM_{2.5}$ concentration in
southeastern BTH, particularly in Beijing and Tianjin, with the $PM_{2.5}$ contribution exceeding
up to 15% (Figure 6b).

It is worth noting that the aerosol species (organic aerosol, sulfate, nitrate, ammonium,

and elemental carbon) exhibit the same variation trend as the $PM_{2.5}$ due to the afforestation
(Figure S5). Apparently, the organic aerosol is the major contributor to the $PM_{2.5}$ variation
due to the afforestation, followed by the sulfate and ammonium aerosol. The afforestation
also increases emissions of the biogenic SOA (BSOA) precursors, such as isoprene and
monoterpenes. However, due to the very low emissions of BSOA precursors during
wintertime (Guenther et al., 2006; 2012), the BSOA contribution to $PM_{2.5}$ concentrations is
insignificant, less than 3 $\mu g\ m^{-3}$ on average during the whole episodes (Figure S6a). The
average BSOA enhancement due to the afforestation is less than 0.5% (Figure S6b).
Furthermore, in general, the afforestation has little effect on the boundary layer height,
upward sensible heat flux (associated with turbulent mixing), and moisture (related to clouds)
in BTH (Figure S7).

To assess the upper limit of impacts of the afforestation on the $PM_{2.5}$ level in BTH, two

additional experiments are conducted and compared to the REF case. The two experiments
are one with complete deforestation and the other with complete afforestation in the Taihang
and Yanshan Mountains (Figures 7a and 7c). In the complete deforestation sensitivity case,
the barren surface with SFz0 of 0.01 m is used to replace other land cover categories. In the
complete afforestation case, the deciduous broadleaf forest category with SFz0 of 0.5 m is
used to replace other land cover categories. As shown in Figure 7, complete deforestation
considerably decreases the $PM_{2.5}$ level in BTH, with the period average $PM_{2.5}$ reduction
ranging from 5~18 $\mu g\ m^{-3}$ generally, and in particular, the $PM_{2.5}$ concentration in Beijing is
reduced by more than 10 $\mu g\ m^{-3}$, due to the intensified northerly or northwesterly wind
caused by the decrease of SFz0 in the Taihang and Yanshan Mountains. Complete
afforestation deteriorates the haze pollution in BTH, and the haze pollution maintains in the
Taihang and Yanshan Mountains due to the weakened northerly or northwesterly wind.
Additionally, the enhancement of $PM_{2.5}$ concentrations in foothill of Taihang and Yanshan
Mountains is obvious, varying from 10 to 25 $\mu g\ m^{-3}$ (Figure 7d).

Interestingly, the afforestation deteriorates most to the haze pollution in Beijing (see

Figure 5). So it is anticipated that the proposed large ventilation corridor system could
alleviate the haze pollution in Beijing (China forest network, 2014, 2016b, c). Originally, the
ventilation corridor system was devised to relieve the urban heat island effect and improve
the thermal environmental conditions in the urbanized regions. With the frequent occurrence
of heavy haze in Beijing, the debatable system is expected to blow away the haze and bring
blue sky to Beijing. In order to examine the effects of the wind corridor system, a sensitivity
experiment is conducted based on the base case, in which three artificial ventilation corridors
are designed in the northwest, north, and northeast of Beijing, with a width of 6 km (Figure
8a). For all the grid cells within the corridors, the barren surface with SFz0 of 0.01 m is used
to replace other land cover categories. Contrast to the anticipation, our sensitivity results
show that the $PM_{2.5}$ reduction due to the designed ventilation corridor system is less than 1%
in Beijing (Figure 8b). Note that the width of the ventilation corridor in the sensitivity study
is 12 times of the proposed one. Hence, the proposed large ventilation corridor system is not
effective or beneficial to mitigate the haze pollution in Beijing.

**4    Summary and conclusions**

The annual land cover product, MCD12Q1, derived from the MODIS observations since

2001 has been used to analyze the land cover change in BTH. A considerable increasing trend
of forests in the western and northwestern BTH has been identified, which is caused by the
China's national afforestation programs. Forests in BTH and Beijing have increased by 7.2%
and 14.9%, respectively, from 2001 to 2013. The fast forest expansion has increased the
surface roughness height, particularly in Beijing and its surrounding areas.

The MCD12Q1 product of 2013 has been assimilated into the WRF-CHEM model to

represent the current land cover condition. Persistent haze pollution episodes in BTH from 1
December 2013 to January 2014 are simulated using the WRF-CHEM model. Generally, the
WRF-CHEM model reasonably well reproduces the temporal variations and spatial
distributions of air pollutants compared to observations at monitoring sites in BTH.

Sensitivity studies have demonstrated that the increase of the surface roughness height

decreases the northwesterly or northerly wind speed in the western and northwestern BTH by
about 0.3~1.5 m s$^{-1}$. The haze pollution is deteriorated in BTH to some degree, and $PM_{2.5}$
concentrations are generally enhanced by less than 6% due to the afforestation. The heavy
haze formation in BTH is generally associated with meteorological conditions when the

northerly or northwesterly wind is weak. Once the northerly or northwesterly wind is strengthened during the haze development in BTH, afforestation plays a considerable role in maintaining high $PM_{2.5}$ concentrations in the downwind of the afforestation area. Complete afforestation or deforestation in the Taihang and Yanshan Mountains would increase or decrease the $PM_{2.5}$ level within 15% in BTH.

Additionally, our model results do not support that the proposed large ventilation corridor system is beneficial to alleviate the haze pollution in Beijing. Under the unfavorable synoptic situations, emissions mitigation is the solely optimum approach to mitigate the haze pollution in BTH.

It is worth to note that, in the present study, contributions of the surface roughness change induced by afforestation to the haze pollution are primarily evaluated using the WRF-CHEM model, but many other factors which directly or indirectly influence air quality, are also modified by the land cover change, including surface moisture, terrestrial erosion, pollutants' dry deposition, PBL thermal stability, etc. For example, changes in surface moisture and surface erosion impact the emissions of natural particles; changes in dry deposition directly influence the air quality in situ and indirectly the air quality downwind with occurrence of recirculation. Therefore, when changes in all those factors caused by land cover change are accounted for, the role of afforestation in air quality in situ might be uncertain. In the online WRF-CHEM model, besides the surface roughness, the impacts of afforestation on the heat flux, surface moisture, surface erosion, and dry deposition of air pollutants have also been considered. Considering that afforestation in BTH is mainly distributed in the mountain region, the surface roughness increase induced by afforestation obviously decrease surface wind speeds, facilitating accumulation of air pollutants in the downwind region and further deteriorating the haze pollution.

## 5    Data availability

The real-time $PM_{2.5}$, $O_3$, $NO_2$, $SO_2$, and CO mass concentrations are accessible to the public on the website http://106.37.208.233:20035/. One can also access the historic profiles of the observed ambient air pollutants by visiting http://www.aqistudy.cn/.

*Acknowledgements.* This work was financially supported by National Key R&D Plan (Quantitative Relationship and Regulation Principle between Regional Oxidation Capacity of Atmospheric and Air Quality (2017YFC0210000)). Long Xin was supported by the Fundamental Research Funds for the Central Universities, and the Project funded by China Postdoctoral Science Foundation (no. 2016M602886) and the Shaanxi Postdoctoral Science Foundation (no. 2017BSHYDZZ27). Guohui Li is supported by "Hundred Talents Program" of the Chinese Academy of Sciences and the National Natural Science Foundation of China (No. 41661144020).

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

Table 1 WRF-CHEM model configurations

| Simulation Regions | Beijing-Tianjin-Hebei-Shandong |
|---|---|
| Simulation period | 1 December 2013 to 31 January 2014 |
| Domain size | 200 × 200 |
| Domain center | 38°N, 116°E |
| Horizontal resolution | 6km × 6km |
| Vertical resolution | 35 vertical levels with a stretched vertical grid with spacing ranging from 30 m near the surface, to 500 m at 2.5 km and 1 km above 14 km |
| Microphysics scheme | WSM 6-class graupel scheme (Hong and Lim, 2006) |
| Boundary layer scheme | MYJ TKE scheme (Janjić, 2002) |
| Surface layer scheme | MYJ surface scheme (Janjić, 2002) |
| Land-surface scheme | Unified Noah land-surface model (Chen and Dudhia, 2001) |
| Longwave radiation scheme | Goddard longwave scheme (Chou and Suarez, 2001) |
| Shortwave radiation scheme | Goddard shortwave scheme (Chou and Suarez, 1999) |
| Meteorological boundary and initial conditions | NCEP 1°×1° reanalysis data |
| Chemical initial and boundary conditions | MOZART 6-hour output (Horowitz et al., 2003) |
| Anthropogenic emission inventory | SAPRC-99 chemical mechanism emissions (Zhang et al., 2009) |
| Biogenic emission inventory | MEGAN model developed by Guenther et al. (2006) |
| Model spin-up time | 28 hours |


Table 2 Land cover change over Beijing and BTH from 2001 to 2013

| Land cover categories | Land cover description | Beijing | BTH |
|---|---|---|---|
| 1~5 | Forests | 14.9% | 7.2% |
| 6~7 | Shrublands | -12.6% | -3.9% |
| 12/14 | Croplands | -0.1 % | 1.9% |
| 8~10 | Grasslands | -2.0% | -5.1% |
| Others | ~ | -0.2% | -0.1% |


**Figure Captions**


Figure 1 (a) The model domain, region of interest (ROI) and monitoring sites. (b) The

topography and monitoring sites in January 2014. The circles represent the centers

of cities with ambient monitoring sites and the size of circles denotes the number of

monitoring sites in the cities. The boundary of BTH region is highlighted with

bright lines. The Yanshan and Taihang Mountains are also displayed.


Figure 2 Land cover change from 2001 to 2013. Spatial distributions of (a) forests, (b)

shrublands, (c) croplands, and (d) grasslands.


Figure 3 Comparisons of observed (black dots) and simulated (solid red lines) diurnal profiles

of near-surface hourly mass concentrations of (a) $PM_{2.5}$, (b) $O_3$, (c) $NO_2$, (d) $SO_2$,

and (d) CO averaged at monitoring sites in BTH from 1 December 2013 to 31

January 2014.


Figure 4 Pattern comparisons of calculated and observed near-surface $PM_{2.5}$ mass

concentrations. (a) Spatial correlation between calculated and observed $PM_{2.5}$

concentrations during each month and the whole simulation period. Horizontal

distribution of calculated (color contour) and observed (colored circles) average

$PM_{2.5}$ concentrations during (b) the whole simulation period, (c) December 2013,

and (d) January 2014, along with the simulated wind fields (black arrows).


Figure 5 (a) SFz0 change from 2001 to 2013, and (b) its correlation with the forest LCF

change; Horizontal distribution of (c) absolute and (d) relative near-surface $PM_{2.5}$

mass concentration changes caused by the afforestation. The wind field changes are

shown in black arrows in (c) and (d).


Figure 6 Horizontal distribution of (a) the average near surface $PM_{2.5}$ mass concentration and

(b) its change due to the afforestation during an intensified northerly/northwesterly

event from 00:00 to 10:00 Beijing Time on January 18, 2014. The wind field and its

change are shown in black arrows.


Figure 7 Horizontal distribution of (a) the average near surface $PM_{2.5}$ mass concentration and

(b) its change due to the afforestation during an intensified northerly/northwesterly

event from 00:00 to 10:00 Beijing Time on January 18, 2014. The wind field and its

change are shown in black arrows.


Figure 8 Impacts of an artificial large ventilation corridor system on (a) SFz0 and (b) average

near-surface $PM_{2.5}$ mass concentrations from 1 December 2013 to 31 January 2014,

along with the wind field (black arrows).






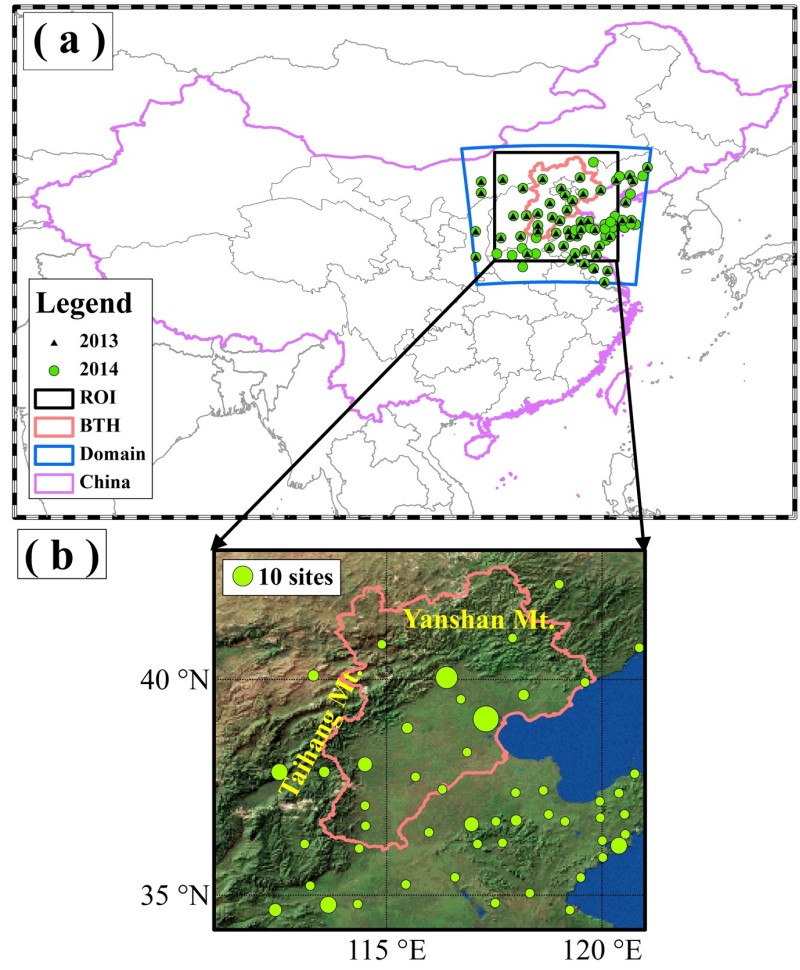

Figure 1 (a) The model domain, region of interest (ROI) and monitoring sites. (b) The
topography and monitoring sites in January 2014. The circles represent the centers of cities
with ambient monitoring sites and the size of circles denotes the number of monitoring sites
in the cities. The boundary of BTH region is highlighted with bright lines. The Yanshan and
Taihang Mountains are also displayed.

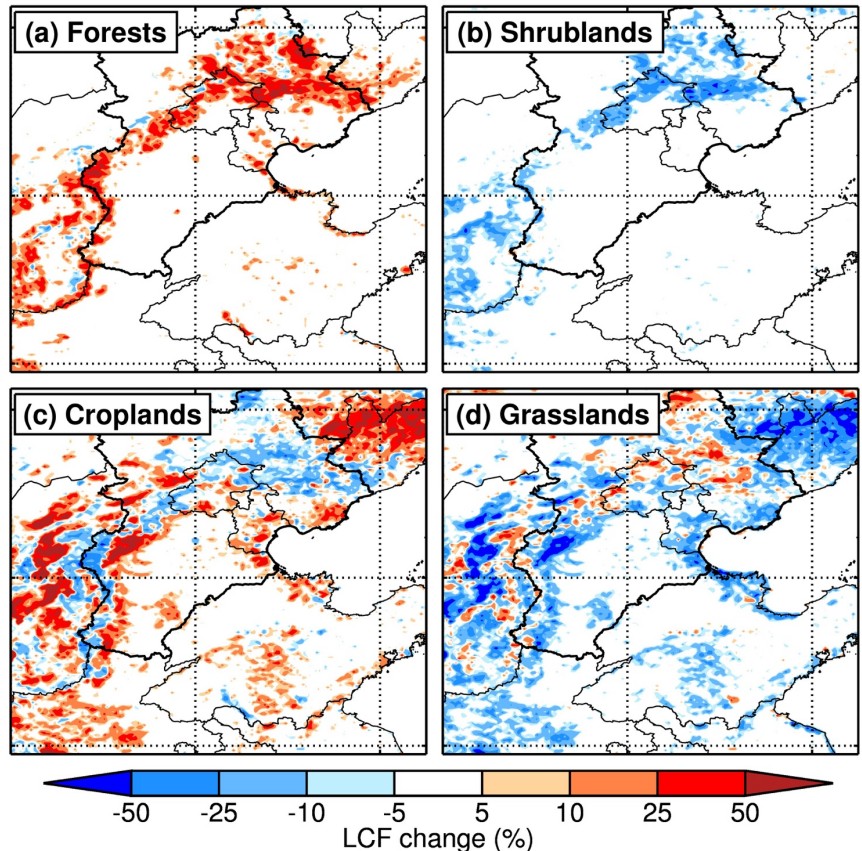

Figure 2 Land cover change from 2001 to 2013. Spatial distributions of (a) forests, (b) shrublands, (c) croplands, and (d) grasslands.

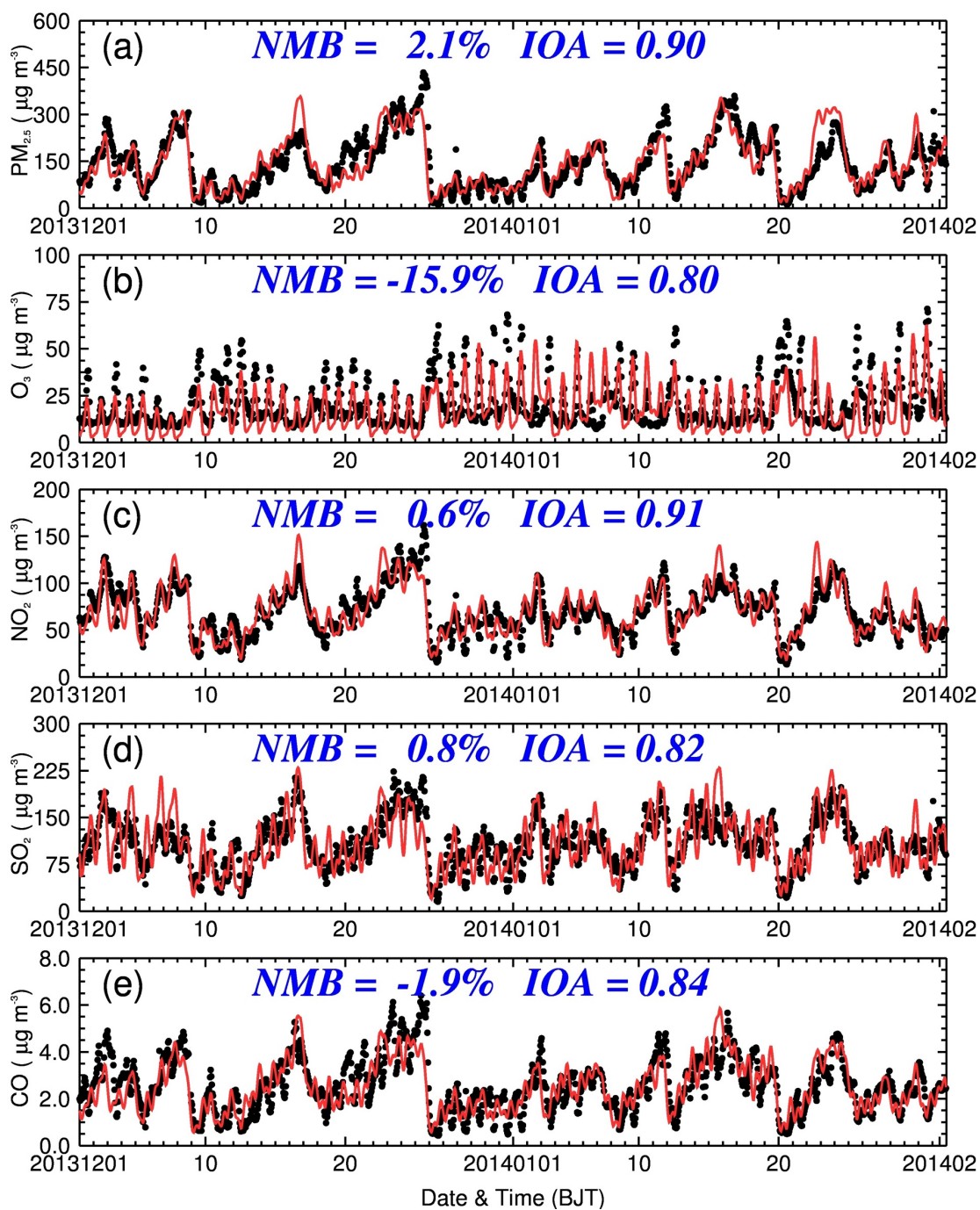

Figure 3 Comparisons of observed (black dots) and simulated (solid red lines) diurnal profiles
of near-surface hourly mass concentrations of (a) PM$_{2.5}$, (b) O$_3$, (c) NO$_2$, (d) SO$_2$, and (d) CO
averaged at monitoring sites in BTH from 1 December 2013 to 31 January 2014.

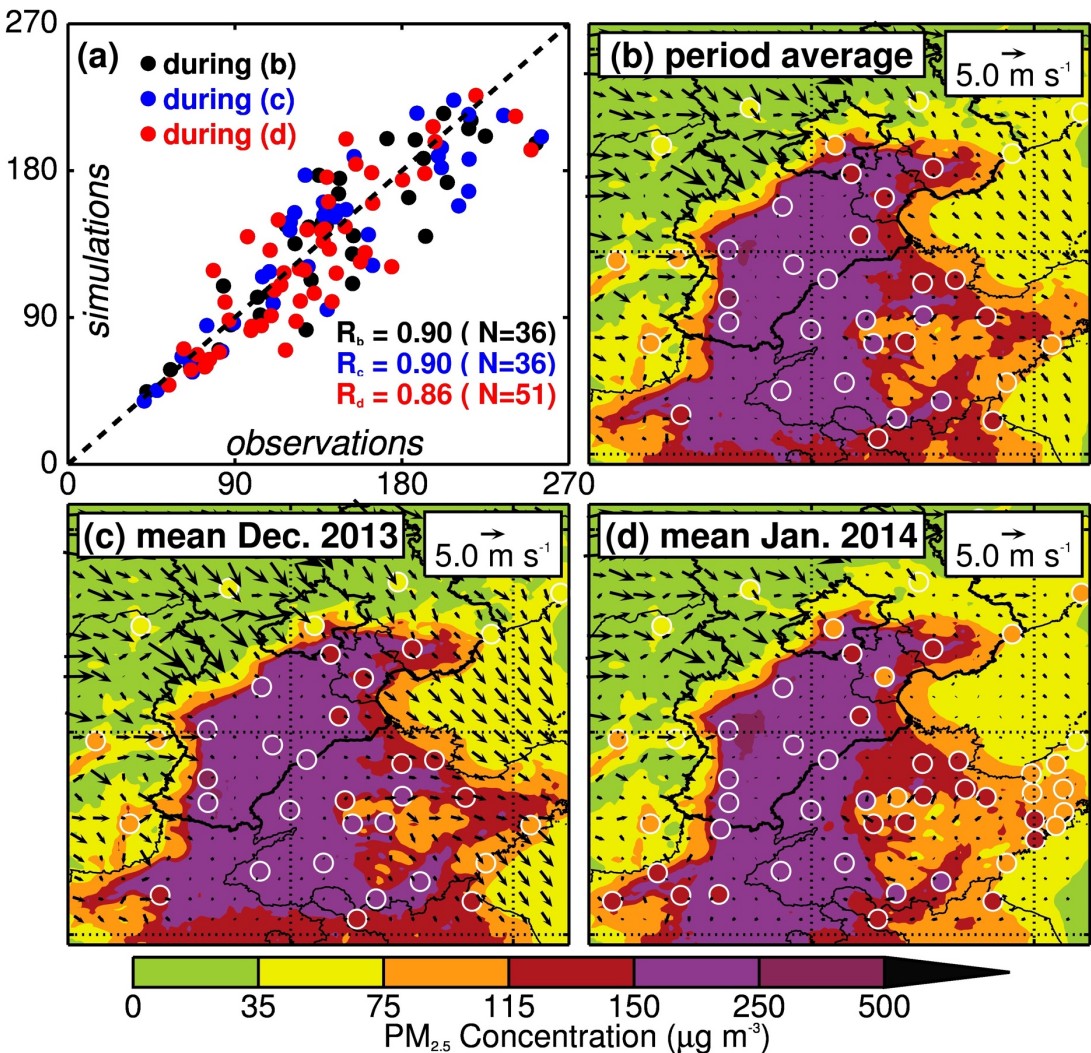

Figure 4 Pattern comparisons of calculated and observed near-surface PM$_{2.5}$ mass concentrations. (a) Spatial correlation between calculated and observed PM$_{2.5}$ concentrations during each month and the whole simulation period. Horizontal distribution of calculated (color contour) and observed (colored circles) average PM$_{2.5}$ concentrations during (b) the whole simulation period, (c) December 2013, and (d) January 2014, along with the simulated wind fields (black arrows).

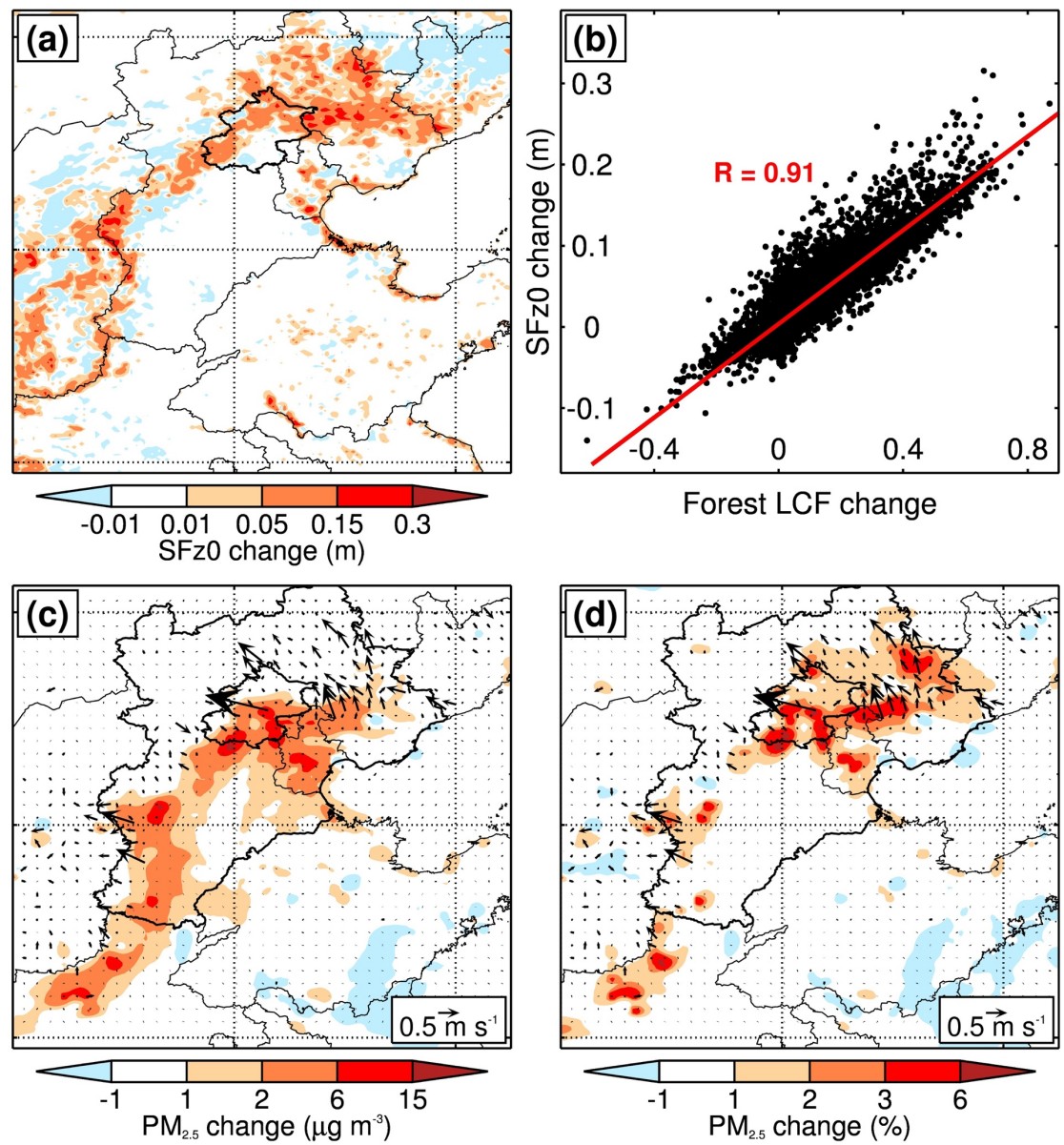

Figure 5 (a) SFz0 change from 2001 to 2013, and (b) its correlation with the forest LCF
change; Horizontal distribution of (c) absolute and (d) relative near-surface $PM_{2.5}$ mass
concentration changes caused by the afforestation. The wind field changes are shown in black
arrows in (c) and (d).

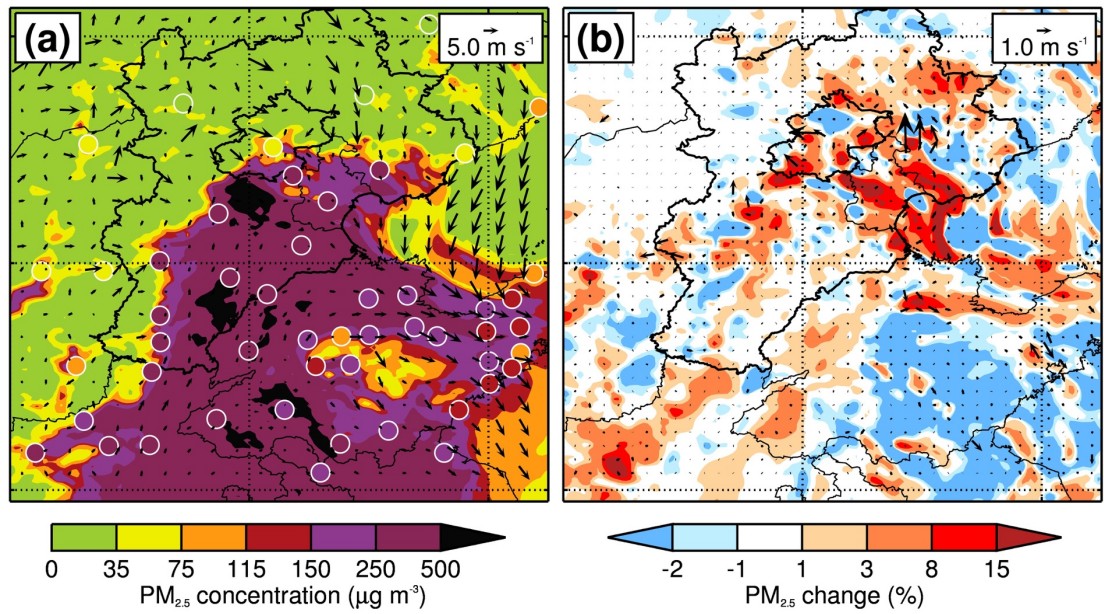

Figure 6 Horizontal distribution of (a) the average near surface PM$_{2.5}$ mass concentration and
(b) its change due to the afforestation during an intensified northerly/northwesterly event
from 00:00 to 10:00 Beijing Time on January 18, 2014. The wind field and its change are
shown in black arrows.

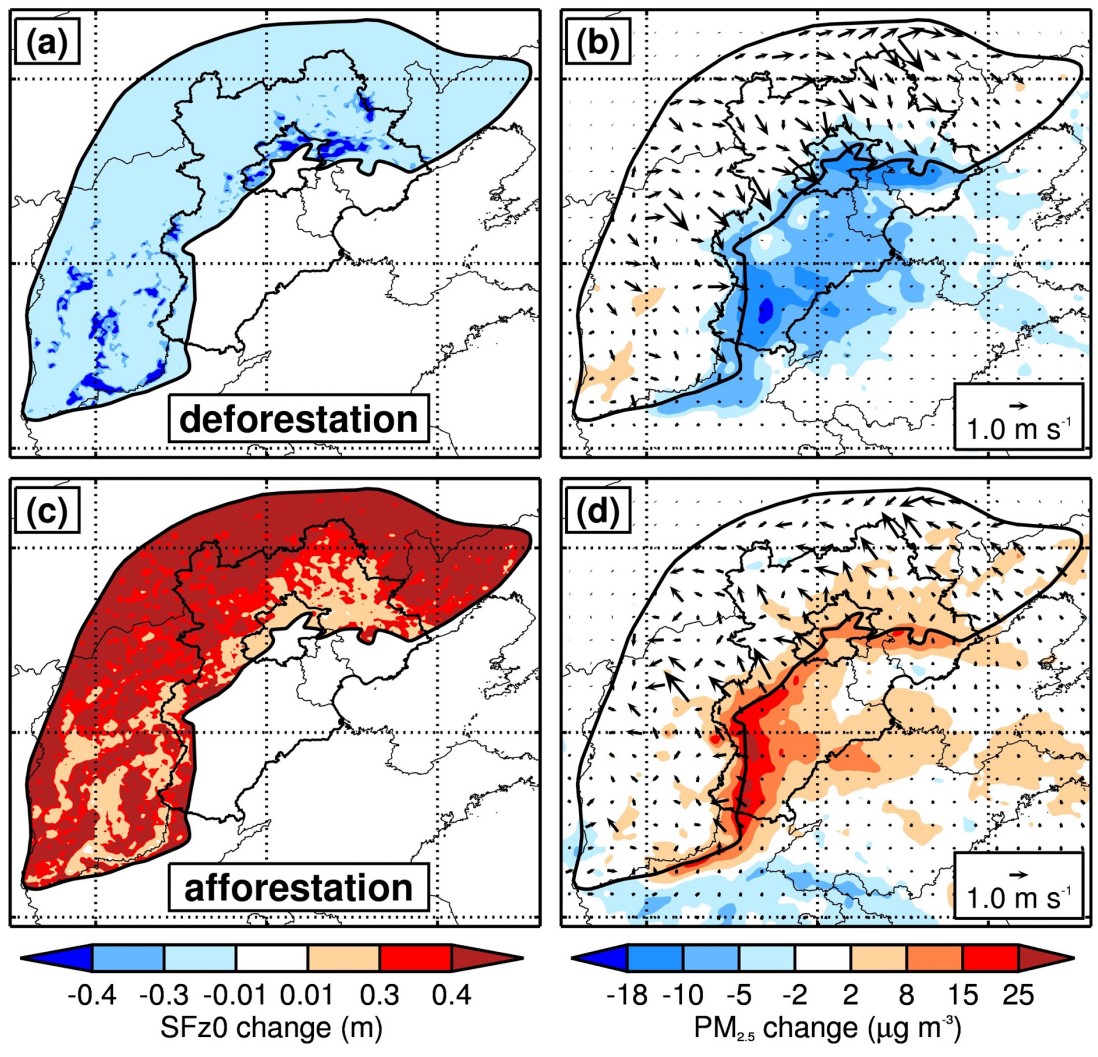

Figure 7 Impacts of complete deforestation/afforestation over Taihang and Yanshan Mountains on (a)/(c) SFz0 and (b)/(d) average near-surface $PM_{2.5}$ mass concentrations from 1 December 2013 to 31 January 2014, along with the wind field change (black arrows).

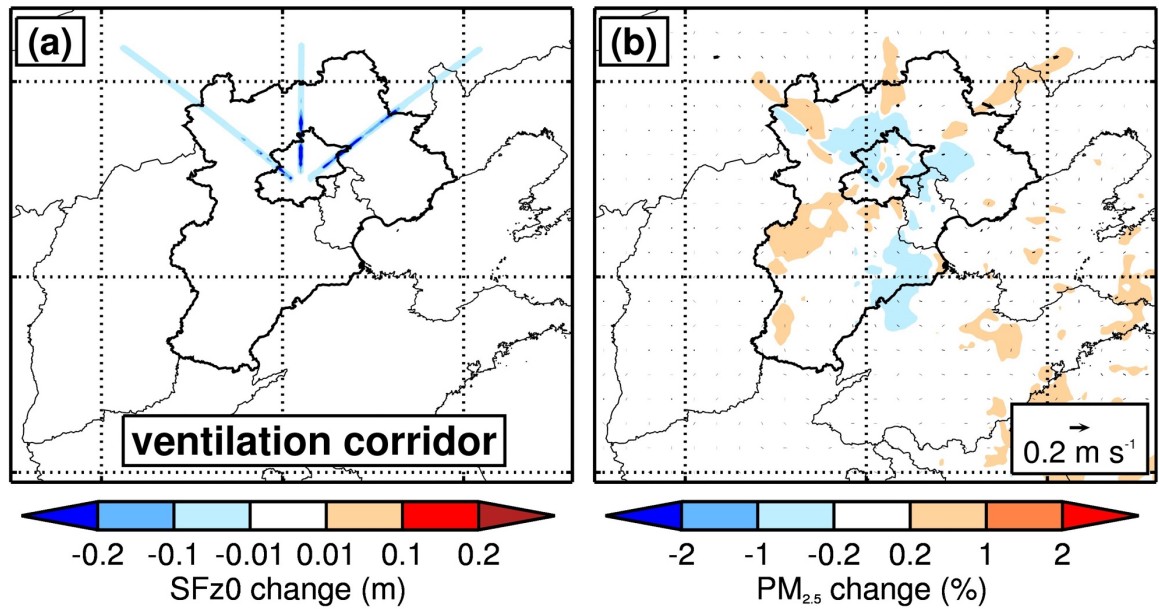

Figure 8 Impacts of an artificial large ventilation corridor system on (a) SFz0 and (b) average
near-surface PM$_{2.5}$ mass concentrations from 1 December 2013 to 31 January 2014, along
with the wind field (black arrows).