# Peer review of "Does afforestation deteriorate haze pollution in Beijing-Tianjin-Hebei (BTH), China?"

_Atmospheric Chemistry and Physics, 2017_

## Short Comment (SC1) · 24 Mar 2018

WRF-Chem modeling was, to some extent, performed using an "improved" roughness length (line 162, eq. 3 which should be eq. 6) which presumably took into consideration of surface heterogeneities within a model grid cell. The influences of surface heterogeneities on surface momentum and heat transfer have been extensively studied in the 1990s partly aiming to more precisely simulate surface stress induced sub-grid inhomogeneous terrain in a climate model. Dynamically, the area average of the roughness length in a heterogeneous terrain would produce the correct spatial average value of the surface stress. A heuristic argument was presented to show that this effective value of z0, or "effective roughness length", could be obtained by averaging drag coefficients based on a 'blending' height approach. In other words, the correct formulation

of an effective roughness length, defined as the area average of the roughness length in heterogeneous terrain, relies upon the appropriate determination of a height scale (blending height). At this height a meteorological quantity is approximately in equilibrium with local surface conditions and independent of horizontal position.

Authors of this paper appeared not aware of the progresses in this aspect. However, they should at least comment on previous studies and compare their model with "effective roughness length" approach, and add some discussions to defend their roughness model which might be subject to uncertainties.

————————————————————

---

## Referee Comment (RC1) · Anonymous Referee #2 · 28 Mar 2018

Review of "Does afforestation deteriorate haze pollution in Beijing-Tianjin-Hebei (BTH), China?" by Long et al.

General comments

The authors report the forest cover change in the BTH during 2001-2013 based on the MODIS product, and they assimilate the land cover change to the WRF-Chem model to investigate the effects of afforestation on haze pollution in this region. Furthermore, they examine whether a speculative and controversial proposal of building a large ventilation corridor system in Beijing would be beneficial in improving the local air quality. The authors conclude that afforestation has minor effects on the haze pollution in BTH, and building the ventilation corridor system would not help in improving the air quality in Beijing either. The manuscript is well presented; I have some minor comments for the authors to address.

Specific comments

1. Land cover change can modify many factors associated with air quality, such as surface roughness, surface moisture and terrestrial erosion, dry deposition of pollutants, thermal stability of PBL, etc. These factors affect air quality directly or indirectly. For example, changing the surface roughness can affect the surface wind speed and consequently affect air quality, upon which this manuscript addresses. A change in surface moisture and surface erosion affect the emissions of natural particles; a change in dry deposition can affect the in situ air quality and the air quality downwind when recirculation occurs. Since this manuscript focuses exclusively on the factor of surface roughness, the authors should clarify this confinement. Other factors could play important roles in improving the air quality, and taking all factors into account, it is likely that afforestation would improve the air quality in the BTH.

2. L230-232, the correlation measures the strength of a linear relationship between two variables; a high correlation coefficient means merely a strong linear relation, but it does not necessarily mean that a variable is a strong contributor of the other one.

---

## Editor Comment (EC1) · Prof. Li (Editor) · 16 May 2018

The two reviewers' comments are so informative and rigorous that I would not seek for any more reviews in order for the authors to respond and proceed to the next stage.

---

## Referee Comment (RC2) · Anonymous Referee #3 · 7 Jun 2018

review to "Does afforestation deteriorate haze pollution in Beijing-Tianjin-Hebei (BTH), China?" by Xin Long et al., MS No.: acp-2017-1239

This is a study of numerical test for model WRF-Chem. Five cases were simulated:(1) the basic case of real land cover in 2013; (2)the case with land cover of 2001; (3)the case with total afforestation; (4) the case with total deforestation; (5) the case with the so called "ventilation corridors" for Beijing. Real emission inventory was used. Results of air pollutants, mainly the PM2.5 concentration, were displayed to shown the influence of land cover change.

The major logic of this paper is: afforestation increases surface roughness, then decrease the wind speed, and then in turn, increase the haze concentration. The numerical experiments support this inference, and give quantitative results, although it is not

significant for the formation of heavy haze in BTH area.

There is a basic problem in above inferring chain. Afforestation increases surface roughness may be true for flat terrain. But in this study of BTH area, afforestation is mainly over mountains (Taihang and Yanshan Mountains). Large uncertainty still exists in parameterization of air - land interaction over complex terrain. In addition to the effect of vegetation, drag of subgrid features of topography should be considered (Jimenez and Dudhia, 2012). Therefore, effect of any change only in land cover (vegetation) may be well within the range of WRF model uncertainty. This paper presents a 6% PM2.5 concentration change for the cases before/after afforestation. It can be regarded as a sensitivity test of the model, rather than a reliable result.

(Reference: Jimenez PA, Dudhia J, 2012, Improving the representation of resolved and unresolved topographic effects on surface wind in the WRF model, Journal of Applied Meteorology & Climatology, 51(2): 300-316.)

My other concerns to this work are:

1)MODIS land cover data, MCD12Q1, was assimilated to the WRF-Chem system. The model performance before and after the data assimilation should be provided.

2)The performance of WRF on representing real meteorological data should be checked, in BTH area.

3)About the simulation case for "ventilation corridors", the width of the corridors is 6km, the horizontal resolution of the model is also 6km. It is hard to resolve this fine structure for the model.

4)How to calculate the wind field difference? Why there is the largest difference of wind in Beijing between the year 2013 and 2001? (Figure 5)

Additional points are:

1)Correspondent to Figure 4, a map of PM source emission is needed.

2)Page 3: "afforestation is beneficial for the atmosphere to remove O3, NOx, SO2, and PM2.5 through the dry deposition process (Zhang et al., 2015; 2017; Huang et al., 2016). Hence, a large artificial ventilation corridor system has been proposed, highly anticipated to ventilate Beijing (China forestry network, 2014, 2016b, c), but why in your corridor experiment the deforestation is used? (Page 12: "In the corridors, the barren surface with SFz0 of 0.01 m is used to replace other land cover categories")

4)Page 4, line 90: "The accuracy of the IGBP layer of MCD12Q1 is estimated to be 72.3-77.4% globally", what about the accuracy in BTH area?

5) Eq (5), Gf should be GT?

6)Page 9, line 211: "The good agreements of the simulated mass concentrations of air pollutants with observations at monitoring sites in BTH show that the emission inventory used in present study and simulated wind fields are generally reasonable". Probably, but not sure. WRF is known for its overestimate of surface wind speeds, which is of importance for air pollution modelling. Here the "good agreements" of haze simulation may imply that other errors in the model have compensated this deficiency.

7) Figure 3, details about the comparison. How the modelled concentration being calculated to compare to the observation? Using the nearest grid point to the observation site?

8)Page 9, line 228-232, "The SFz0 change is highly correlated with the forest LCF change, with a correlation coefficient of 0.91, indicating that the afforestation is the most important factor for the increase in the SFz0 in BTH." This is just expected results! Need not to be "indicating".

9)Page 10, line 234, "The prevailing wind is decelerated...", what do prevailing wind mean here?

10)Page 10, 239, " The PM2.5 enhancement in Beijing is the most evident, corresponding to the rapid growth of forests in the west and in/on the north of Beijing". This

СЗ

is doubtful. How can the air pollution so sensitive to local change of land cover?

---

## Author Comment (AC1) · 19 Jul 2018

**Reply to Dr. Ma**

We thank the reviewer for the careful reading of the manuscript and helpful comments. We have revised the manuscript following the suggestion, as described below.

**Comment:** WRF-Chem modeling was, to some extent, performed using an "improved" roughness length (line 162, eq. 3 which should be eq. 6), which presumably took into consideration of surface heterogeneities within a model grid cell. The influences of surface heterogeneities on surface momentum and heat transfer have been extensively studied in the 1990s partly aiming to more precisely simulate surface stress induced sub-grid inhomogeneous terrain in a climate model. Dynamically, the area average of the roughness length in a heterogeneous terrain would produce the correct spatial average value of the surface stress. A heuristic argument was presented to show that this effective value of z0, or "effective roughness length", could be obtained by averaging drag coefficients based on a 'blending' height approach. In other words, the correct formulation of an effective roughness length, defined as the area average of the roughness length in heterogeneous terrain, relies upon the appropriate determination of a height scale (blending height). At this height a meteorological quantity is approximately in equilibrium with local surface conditions and independent of horizontal position. Authors of this paper appeared not aware of the progresses in this aspect. However, they should at least comment on previous studies and compare their model with "effective roughness length" approach, and add some discussions to defend their roughness model, which might be subject to uncertainties.

**Response**:
We have revised the Eq. 3 to Eq. 6 in Section 2.3.
We have also clarified in Section 2.3: "*In order to more precisely simulate surface stress within the sub-grid scale in heterogeneous terrain, the effective roughness length has been extensively studied, especially in the 1990s. Claussen (1990) has defined the effective roughness length as a value of the area average of the roughness length in heterogeneous terrain. The effective roughness length relies upon the*

*blending height (Wieringa, 1986; Mason, 1988; Wood and Mason, 1991; Philip, 1996; Mahrt, 1996), at which the flow is approximately in equilibrium with underlying surface conditions and independent of horizontal position (Ma and Daggupary, 1998). We have modified the Noah SFz0 calculation using the spatial average of the vegetation roughness length.*"

**References:**

Claussen, M.: Area-averaging of surface fluxes in a neutrally stratified, horizontally inhomogeneous atmospheric boundary layer, Atmos. Environ., 24, 1349-1360, 1990.

Ma, J., and Daggupaty S. M.: Stability Dependence of Height Scales and Effective Roughness Lengths of Momentum and Heat Transfer Over Roughness Changes, Bound. Lay. Meteorol., 88, 145-160, 1998.

Mahrt, L.: The bulk aerodynamic formulation over heterogeneous surfaces, Bound.-Lay. Meteorol., 78, 87-119, 1996.

Mason, P. J.: The formation of area-averaged roughness lengths. Quart. J. Roy. Meteorol. Soc., 114, 399-420, 1988.

Philip, J. R.: Two-dimensional checkerboards and blending heights, Bound. Lay. Meteorol., 80, 1-18, 1996.

Wieringa, J.: Roughness-dependent geographical interpolation of surface wind speed averages, Quart. J. Roy. Meteorol. Soc., 112, 867-889, 1986.

Wood, N., and Mason, P.: The influence of static stability on the effective roughness lengths for momentum and heat transfer, Quart. J. Roy. Meteorol. Soc., 117, 1025-1056, 1991.

---

## Author Comment (AC2) · 19 Jul 2018

Dear Editor,

We have received the comments from the two reviewers and Dr. Ma of the manuscript. Below are our responses and the revisions that we have made in the manuscript.

Thank you for your efforts on this manuscript. We look forward to hearing from you.

Best Regards,

Guohui Li

---

## Author Comment (AC3) · 19 Jul 2018

**Response to Referee #2**

We thank the reviewer for the careful reading of the manuscript and helpful comments. We have revised the manuscript following the suggestion, as described below.

**General comments**

The authors report the forest cover change in the BTH during 2001-2013 based on the MODIS product, and they assimilate the land cover change to the WRF-Chem model to investigate the effects of afforestation on haze pollution in this region. Furthermore, they examine whether a speculative and controversial proposal of building a large ventilation corridor system in Beijing would be beneficial in improving the local air quality. The authors conclude that afforestation has minor effects on the haze pollution in BTH, and building the ventilation corridor system would not help in improving the air quality in Beijing either. The manuscript is well presented; I have some minor comments for the authors to address.

**Special comments**

**Comment:** Land cover change can modify many factors associated with air quality, such as surface roughness, surface moisture and terrestrial erosion, dry deposition of pollutants, thermal stability of PBL, etc. These factors affect air quality directly or indirectly. For example, changing the surface roughness can affect the surface wind speed and consequently affect air quality, upon which this manuscript addresses. A change in surface moisture and surface erosion affect the emissions of natural particles; a change in dry deposition can affect the in situ air quality and the air quality downwind when recirculation occurs. Since this manuscript focuses exclusively on the factor of surface roughness, the authors should clarify this confinement. Other factors could play important roles in improving the air quality, and taking all factors into account, it is likely that afforestation would improve the air quality in the BTH.

**Response**: We have clarified in Section 4: "It is worth to note that, in the present study, contributions of the surface roughness change induced by afforestation to the

haze pollution are primarily evaluated using the WRF-CHEM model, but many other factors which directly or indirectly influence air quality, are also modified by the land cover change, including surface moisture, terrestrial erosion, pollutants' dry deposition, PBL thermal stability, etc. For example, changes in surface moisture and surface erosion impact the emissions of natural particles; changes in dry deposition directly influence the air quality in situ and indirectly the air quality downwind with occurrence of recirculation. Therefore, when changes in all those factors caused by land cover change are accounted for, the role of afforestation in air quality in situ might be uncertain. In the online WRF-CHEM model, besides the surface roughness, the impacts of afforestation on the heat flux, surface moisture, surface erosion, and dry deposition of air pollutants have also been considered. Considering that afforestation in BTH is mainly distributed in the mountain region, the surface roughness increase induced by afforestation obviously decrease surface wind speeds, facilitating accumulation of air pollutants in the downwind region and further deteriorating the haze pollution."

**Comment:** L230-232, the correlation measures the strength of a linear relationship between two variables; a high correlation coefficient means merely a strong linear relation, but it does not necessarily mean that a variable is a strong contributor of the other one.

**Response**: We agree with the reviewer's comment, and have clarified in Section 3.3: "The SFz0 change is highly correlated with the forest LCF change, with a correlation coefficient of 0.91. Generally, the SFz0 is mainly dependent upon the LCF (Equation 6), and sensitive to the forest change (Table S2). Therefore, afforestation constitutes the most important factor for the increase in the SFz0 in BTH."

---

## Author Comment (AC4) · 19 Jul 2018

**Reply to Anonymous Referee #3**

We thank the reviewer for the careful reading of the manuscript and helpful comments. We have revised the manuscript following the suggestion, as described below.

**General comments**

**Comment:** This is a study of numerical test for model WRF-Chem. Five cases were simulated:(1) the basic case of real land cover in 2013; (2) the case with land cover of 2001; (3) the case with total afforestation; (4) the case with total deforestation; (5) the case with the so called "ventilation corridors" for Beijing. Real emission inventory was used. Results of air pollutants, mainly the $PM_{2.5}$ concentration, were displayed to shown the influence of land cover change.

The major logic of this paper is: afforestation increases surface roughness, then decrease the wind speed, and then in turn, increase the haze concentration. The numerical experiments support this inference, and give quantitative results, although it is not significant for the formation of heavy haze in BTH area.

There is a basic problem in above inferring chain. Afforestation increases surface roughness may be true for flat terrain. But in this study of BTH area, afforestation is mainly over mountains (Taihang and Yanshan Mountains). Large uncertainty still exists in parameterization of air - land interaction over complex terrain. In addition to the effect of vegetation, drag of subgrid features of topography should be considered (Jimenez and Dudhia, 2012). Therefore, effect of any change only in land cover (vegetation) may be well within the range of WRF model uncertainties. This paper presents a 6% $PM_{2.5}$ concentration change for the cases before/after afforestation. It can be regarded as a sensitivity test of the model, rather than a reliable result. (Reference: Jimenez PA, Dudhia J, 2012, Improving the representation of resolved and unresolved topographic effects on surface wind in the WRF model, Journal of Applied Meteorology & Climatology, 51(2): 300-316.)

**Response:** We have clarified in Section 3.3: "*It is worth noting that Jiménez and Dudhia (2012) have point out that there still exist large uncertainties in parameterizing the air land interaction over complex terrain. Besides the vegetation*

*effect on the roughness length, drag of subgrid features of topography need to be considered. The parameterization of orographic flow over complex terrain is a challengeable problem at the mesoscale numerical simulation. In early versions of the WRF model, a large bias in predicting surface winds over complex terrain has occurred due to the drag exerted by unresolved topography (Cheng and Steenburgh, 2005). Great efforts have been made to improve the simulation of orographic flow over complex terrain. The new parameterization scheme introduced in the WRF model since version 3.4.1 has corrected this high wind speed bias over plains and valleys (Mughal et al., 2017), and also corrected the low wind speed bias found over the mountains and hills (Jiménez and Dudhia, 2012).*"

Furthermore, we have clarified in Section 2.3: "*In order to more precisely simulate surface stress within the sub-grid scale in heterogeneous terrain, the effective roughness length has been extensively studied, especially in the 1990s. Claussen (1990) has defined the effective roughness length as a value of the area average of the roughness length in the heterogeneous terrain. The effective roughness length relies upon the blending height (Wieringa 1986; Mason 1988; Wood and Mason 1991; Philip 1996; Mahrt 1996), at which the flow is approximately in equilibrium with underlying surface conditions and independent of horizontal position (Ma and Daggupary 1998). We have modified the Noah SFz0 calculation using the spatial average of the vegetation roughness length.*".

**Other concerns**

**Comment (1):** MODIS land cover data, MCD12Q1, was assimilated to the WRF-Chem system. The model performance before and after the data assimilation should be provided.

**Response**: We have provided the validation of $PM_{2.5}$, $O_3$, $NO_2$, $SO_2$, and CO using the measurements at monitoring sites in BTH, and compared the simulated wind speed and direction, and planetary boundary layer height at monitoring sites with the reanalysis data from the European Centre for Medium-range Weather Forecasts (ECMWF) for the REF case (after the data assimilation). Comparisons have shown that the difference of the simulated air pollutants and meteorological parameters

between the SEN-AFF case (before the data assimilation) and the REF case is not significant, so the model performance for the SEN-AFF case is not provided further in the manuscript. We have clarified in Section 3.3: "*On average, the difference of the simulated air pollutants and meteorological parameters between the REF and SEN-AFF case is not significant.*". In addition, please refer to **Comment (2)** about the validation for the meteorological parameters.

**Comment (2):** The performance of WRF on representing real meteorological data should be checked, in BTH area.

**Response**: We have clarified in Section 3.2: "*Considering the key role of meteorological fields in determining the formation, transformation, diffusion, transport, and removal of the air pollutants (Bei et al., 2017), Figure S2 presents the comparison of the simulated wind speed and direction, and planetary boundary layer height with the reanalysis data from ECMWF (European Centre for Medium-range Weather Forecasts) at monitoring sites. The predicted temporal variations of the three meteorological parameters are generally in agreement with the reanalysis data, with the IOAs exceeding 0.80, and the absolute NMB less than 25%.*"

**Comment (3):** About the simulation case for "ventilation corridors", the width of the corridors is 6 km, the horizontal resolution of the model is also 6 km. It is hard to resolve this fine structure for the model.

**Response**: We have clarified in detail in Section 3.3: "*For all the grid cells within the corridors, the barren surface with SFz0 of 0.01 m is used to replace other land cover categories.*"

**Comment (4):** How to calculate the wind field difference? Why there is the largest difference of wind in Beijing between the year 2013 and 2001? (Figure 5)

**Response**: We calculate the wind field difference by subtracting the simulated U and V components in SEN-AFF case from those in the REF case: U(REF) – U(SEN-AFF)

and V(REF) – V(SEN-AFF). The largest difference of winds in Beijing between the year 2013 and 2001 is caused by the rapid growth of forests in the west and in/on the north of Beijing compared to other regions in BTH, which causes the slowdown of the westerly or northerly wind appreciably.

**Additional points**

**Comment:** Correspondent to Figure 4, a map of PM source emission is needed.

**Response**: We have included the emission distribution of OC, VOCs, $NO_x$, and $SO_2$ in Figure S1. We have clarified in Section 2.2: "*Figure S1 shows the emission distribution of OC, VOCs, NO$_x$, and SO$_2$ in the simulation domain. The high emissions of OC, VOCs, NOx, and SO$_2$ are generally concentrated in the plain region of BTH and Shandong province, the downwind area of afforestation.*"

**Comment:** Page 3: "afforestation is beneficial for the atmosphere to remove $O_3$, NOx, SO2, and $PM_{2.5}$ through the dry deposition process (Zhang et al., 2015; 2017; Huang et al., 2016). Hence, a large artificial ventilation corridor system has been proposed, highly anticipated to ventilate Beijing (China forestry network, 2014, 2016b, c), but why in your corridor experiment the deforestation is used? (Page 12: " In the corridors, the barren surface with SFz0 of 0.01 m is used to replace other land cover categories")

**Response**: We do not use the deforestation in the corridor. It is supposed that all the vegetation and buildings should be removed in the proposed corridor, so we use the barren surface with SFz0 of 0.01 m in the corridor.

**Comment:** Page 4, line 90: "The accuracy of the IGBP layer of MCD12Q1 is estimated to be 72.3-77.4% globally", what about the accuracy in BTH area?

**Response**: We have clarified in Section 2.1: "*Great efforts have been made to evaluate the accuracies of the global land cover datasets over China. The overall accuracy of MCD12Q1 in China is estimated to be 55.9-68.9% (Bai et al., 2015;*

*Yang et al., 2017), which could be increased to about 70% when ignoring the differences of five forests.*"

**Comment:** Eq (5), Gf should be GT?

**Response**: $G_f$ is not $G_T$, and we have clarified in Section 2.3: "*$G_f$ is the area fractional coverage of green vegetation, and $G_T$, $G_{min}$ and $G_{max}$ are the threshold, minimal, and maximal value of $G_f$, respectively.*"

**Comment:** Page 9, line 211: "The good agreements of the simulated mass concentrations of air pollutants with observations at monitoring sites in BTH show that the emission inventory used in present study and simulated wind fields are generally reasonable". Probably, but not sure. WRF is known for its overestimate of surface wind speeds, which is of importance for air pollution modeling. Here the "good agreements" of haze simulation may imply that other errors in the model have compensated this deficiency.

**Response**: We have provided validation for the meteorological parameters, including wind speed and direction, and planetary boundary layer height, and the predicted meteorological parameters are generally consistent with the reanalysis data. We have clarified in Section 3.2: "*It is worth noting that, although the predicted meteorological parameters are generally consistent with the reanalysis data from ECMWF at monitoring sites, other factors still affect the meteorological field simulations and cause biases to compensate some of the deficiencies of the WRF-CHEM model, such as overestimation of surface wind speeds.*"

**Comment:** Figure 3, details about the comparison. How the modelled concentration being calculated to compare to the observation? Using the nearest grid point to the observation site?

**Response**: Yes, we use the simulation of nearest gird point to the observation site to compare with the measurement.

**Comment:** Page 9, line 228-232, "The SFz0 change is highly correlated with the forest LCF change, with a correlation coefficient of 0.91, indicating that the afforestation is the most important factor for the increase in the SFz0 in BTH." This is just expected results! Need not to be "indicating".

**Response**: We have clarified in Section 3.3: "*The SFz0 change is highly correlated with the forest LCF change, with a correlation coefficient of 0.91. Generally, the SFz0 is mainly dependent upon the LCF (Equation 6), and sensitive to the forest change (Table S2). Therefore, afforestation constitutes the most important factor for the increase in the SFz0 in BTH.*"

**Comment:** Page 10, line 234, "The prevailing wind is decelerated...", what do prevailing wind mean here?

**Response**: We revised the sentence in Section 3.3: "*The prevailing westerly or northerly wind is decelerated...*"

**Comment:** Page 10, 239, "The PM$_{2.5}$ enhancement in Beijing is the most evident, corresponding to the rapid growth of forests in the west and in/on the north of Beijing". This is doubtful. How can the air pollution so sensitive to local change of land cover?

**Response**: We have concluded that afforestation does not play an important role in the haze pollution in BTH, enhancing PM$_{2.5}$ concentrations by up to 6% on average. Compared to other regions in BTH, the rapid growth of forests in the west and in/on the north of Beijing cause the slowdown of the westerly or northerly wind appreciably, unfavorable for dispersion of air pollutants. Therefore, the PM$_{2.5}$ enhancement in Beijing caused by afforestation is generally the most obvious.

**References**

Bai, Y., Feng, M., Jiang, H., Wang, J., and Liu, Y.: Validation of land cover maps in China using a sampling-based labeling approach, Remote SENS-BASEL, 7, 10589-10606, 2015.

Cheng, W. Y. Y., and Steenburgh, W. J.: Evaluation of surface sensible weather forecasts by

the WRF and the Eta Models over the western United States, Weather Forecast., 20, 812-821, 10.1175/waf885.1, 2005.

Jiménez, P. A., and Dudhia, J.: Improving the representation of resolved and unresolved topographic effects on surface wind in the WRF Model, J. Appl. Meteorol. Clim., 51, 300-316, 2012.

Mughal, M. O., Lynch, M., Yu, F., Mcgann, B., Sutton, J., and Sutton, J.: Wind modelling, validation and sensitivity study using Weather Research and Forecasting model in complex terrain, Environ. Modell. Softw., 90, 107-125, 2017.

Yang, Y., Xiao, P., Feng, X., and Li, H.: Accuracy assessment of seven global land cover datasets over China, Isprs J. Photogramm., 125, 156-173, 2017.